# Polo-like Kinase 1 Predicts Lymph Node Metastasis in Middle Eastern Colorectal Cancer Patients; Its Inhibition Reverses 5-Fu Resistance in Colorectal Cancer Cells

**DOI:** 10.3390/cells13201700

**Published:** 2024-10-14

**Authors:** Pratheesh Kumar Poyil, Abdul K. Siraj, Divya Padmaja, Sandeep Kumar Parvathareddy, Khadija Alobaisi, Saravanan Thangavel, Rafia Begum, Roxanne Diaz, Fouad Al-Dayel, Khawla S. Al-Kuraya

**Affiliations:** 1Human Cancer Genomic Research, King Faisal Specialist Hospital and Research Center, P.O. Box 3354, Riyadh 11211, Saudi Arabia; ppoyil@kfshrc.edu.sa (P.K.P.); asiraj@kfshrc.edu.sa (A.K.S.); dvsasidharan50@gmail.com (D.P.); psandeepkumar@kfshrc.edu.sa (S.K.P.); kalobaisi@kfshrc.edu.sa (K.A.); tsaravanan97@kfshrc.edu.sa (S.T.); brafia@kfshrc.edu.sa (R.B.); rmelosantos87@kfshrc.edu.sa (R.D.); 2Department of Pathology, King Faisal Specialist Hospital and Research Center, P.O. Box 3354, Riyadh 11211, Saudi Arabia; dayelf@kfshrc.edu.sa

**Keywords:** PLK1, CRC, stemness, EMT, Zeb1

## Abstract

Polo-like kinase 1 (PLK1) is a serine/threonine–protein kinase essential for regulating multiple stages of cell cycle progression in mammals. Aberrant regulation of PLK1 has been observed in numerous human cancers and is linked to poor prognoses. However, its role in the pathogenesis of colorectal cancer (CRC) in the Middle East remains unexplored. PLK1 overexpression was noted in 60.3% (693/1149) of CRC cases and was significantly associated with aggressive clinico-pathological parameters and p-ERK1/2 overexpression. Intriguingly, multivariate logistic regression analysis identified PLK1 as an independent predictor of lymph node metastasis. Our in vitro experiments demonstrated that CRC cells with high PLK1 levels were resistant to 5-Fu treatment, while those with low PLK1 expression were sensitive. To investigate PLK1′s role in chemoresistance, we used the specific inhibitor volasertib, which effectively reversed 5-Fu resistance. Interestingly, forced PLK1 expression activated the CRAF-MEK-ERK signaling cascade, while its inhibition suppressed this cascade. PLK1 knockdown reduced epithelial-to-mesenchymal transition (EMT) progression and stem cell-like traits in 5-Fu-resistant cells, implicating PLK1 in EMT induction and stemness in CRC. Moreover, silencing ERK1/2 significantly mitigated chemoresistance, EMT, and stemness properties in CRC cell lines that express PLK1. Furthermore, the knockdown of Zeb1 attenuated EMT and stemness, suggesting a possible link between EMT activation and the maintenance of stemness in CRC. Our findings underscore the pivotal role of PLK1 in mediating chemoresistance and suggest that PLK1 inhibition may represent a potential therapeutic strategy for the management of aggressive colorectal cancer subtypes.

## 1. Introduction

Colorectal cancer (CRC) is among the most predominant cancers globally and is the second leading cause of cancer-related deaths in the United States [1,2]. In Saudi Arabia, the incidence of CRC is on the rise; CRC is the most common cancer affecting Saudi males [3].

Although numerous new therapeutic agents have been developed, 5-Fluorouracil (5-Fu)-based combination chemotherapy continues to be the standard treatment for CRC patients. However, the overall response rate to 5-Fu remains below 15%, largely due to the development of intrinsic or acquired chemoresistance upon 5-Fu exposure, which significantly limits its clinical efficacy [4]. The emergence of chemoresistance has contributed to the development of liver metastasis in approximately 50% of CRC patients during disease progression. This greatly influences treatment outcomes and increases the complexity of disease management [5,6].

Advanced CRC is associated with high mortality rates, primarily due to its propensity to metastasize to the lymph nodes, peritoneum, lungs, and liver [7]. Tumor metastasis involves a complex interplay of cellular and molecular processes, with epithelial–mesenchymal transition (EMT) serving as a critical early step in the progression to metastasis [7]. EMT is a biological process in which epithelial cells undergo a phenotypic transformation, losing their defining characteristics and acquiring properties associated with mesenchymal cells. While EMT is essential for early developmental processes such as embryogenesis, tissue formation, and fibrosis, it has also been linked to cancer progression. Specifically, EMT contributes to tumor growth, cell proliferation, drug resistance, and metastasis [8].

Human Polo-like kinase 1 (PLK1) has recently been identified as a critical driver of the EMT process [9]. PLK1 is a key mitotic protein kinase that plays a critical role in cell division and serves as a major regulator of cellular proliferation, making it an attractive therapeutic target for cancer treatment [10,11,12,13,14]. PLK1 is suggested to be a key driver of EMT and metastasis in prostate cancer [15], gastric cancer [16], lung cancer [17], pancreatic ductal adenocarcinoma [18], and renal cell carcinoma [19]. PLK1 is known to be overexpressed in numerous tumors relative to their normal tissue counterparts, and this overexpression has been associated with poor prognosis [11,20,21]. Additionally, its overexpression has been linked to therapy resistance, while its inhibition has been shown to restore sensitivity to chemotherapy [22]. Previous studies have shown that PLK1 is overexpressed in CRC and is associated with disease progression, metastasis, and poor prognosis [23,24,25], while its precise role in these processes remains under investigation.

Furthermore, the clinicopathological significance and prognostic value of PLK1 in Middle Eastern CRC patients have yet to be investigated, suggesting the clinical demand to further investigate PLK1 dysregulation in CRC from different ethnicities and to investigate the potential mechanism of PLK1 inhibitors in CRC.

This study demonstrated that the dysregulated activation of PLK1 signaling serves as a predictive prognostic biomarker for lymph node metastasis. In vitro experiments show that PLK1 facilitated EMT by activating the MEK/ERK signaling pathway, and inhibition of PLK1 overcame resistance to 5-Fu in CRC cells. Thus, PLK1 may serve as a promising therapeutic target for metastatic or refractory CRC.

## 2. Materials and Methods

### 2.1. Sample Selection

One thousand one-hundred and forty-nine patients diagnosed CRC between 1990 and 2018 at King Faisal Specialist Hospital and Research Center in Riyadh, Saudi Arabia, were included in the study. Clinicopathological data were derived from patient records and included age, sex, tumor site, histological subtype, stage, and grade of the tumor. They are summarized in Table 1. All samples were obtained from archival tissues and, hence, waiver of consent was granted by the Research Advisory Council (RAC) of the hospital’s Institutional Review Board under project RAC# 2190016.

### 2.2. Tissue Microarray Construction and Immunohistochemistry 

All samples were analyzed using a tissue microarray (TMA) format. For the construction of TMA, tissue cylinders with a diameter of 0.6 mm were punched from representative tumor regions of each donor tissue block and inserted into a recipient paraffin block using a modified semiautomatic robotic precision instrument (Beecher Instruments, Woodland, WI, USA). Two CRC cores were arrayed from each case.

Standard protocol was followed for immunohistochemical staining. Antigen retrieval was performed using Dako (Dako Denmark A/S, Glostrup, Denmark) Target Retrieval Solution pH 9.0 (Catalog number S2367), with the slides being placed in a Pascal pressure cooker (Agilent technologies, Glostrup, Denmark) at a temperature of 1200C for 8 min. Following antigen retrieval, the slides were incubated overnight with primary antibodies against PLK-1 (mouse monoclonal, ab-17056, 1:500, pH 9.0; Abcam, Cambridge, United Kingdom) and p-ERK 1/2 (cat. no. 4376S, 1:1000, pH 6.0; Cell Signaling Technology, Danvers, MA, USA). Secondary detection was performed using the Dako EnVision Plus System kit (Agilent technologies, Glostrup, Denmark), and 3,3′-diaminobenzidine (Agilent technologies, Glostrup, Denmark) was used as the chromogen. Counterstaining was performed using hematoxylin, followed by dehydration, clearing, and mounting of the slides. Negative controls were prepared by omitting the primary antibody. Normal tissues from various organ systems were included in the TMA as controls. To reduce the impact of slide aging and ensure experiment reproducibility, only freshly cut slides were stained simultaneously.

PLK-1 staining was scored using immunoreactivity score (IRS), as previously described [26]. Staining intensity was scored as negative (score 0), weak (score 1), moderate (score 2), strong (score 3). The proportion of positively stained tumor cells was scored as follows: score 0 = 0%; score 1 = 1–10%; score 2 = 11–50%; score 3 = 51–80%; and score 4 = 80–100% positively stained tumor cells. The final IRS score was determined by multiplying the intensity and proportion scores (range: 0–12). PLK1 expression was classified as low with an IRS of 0–6 and high if the IRS was greater than 6. p-ERK 1/2 staining was scored using H score. The staining intensity was scored as described above, and the proportion of tumor staining for that intensity was recorded as 5% increments ranging from 0 to 100%. A final H score (range 0–300) was obtained by adding the sum of scores obtained for each intensity and proportion of area stained. X-tile software (version 3.6.1) [27] was used to define the optimal cut-off for the H score. Based on X-tile plots, PTC cases with H score ≤ 60 were defined as low expression, and the other group (H score > 60) was defined as overexpression for p-ERK 1/2. Mismatch repair (MMR) protein staining was performed for MLH1, MSH2, MSH6, and PMS2, with complete absence of nuclear staining in the tumor cells being classified as deficient MMR and presence of any nuclear staining classified as proficient MMR.

### 2.3. Tissue Culture Experiments

Human colorectal cancer (CRC) cell lines—HT29, HCT116, DLD1, and SW480—were obtained from the American Type Culture Collection (ATCC, Manassas, VA, USA). The cell lines were cultured in RPMI 1640 medium supplemented with 10% fetal bovine serum (FBS, ATCC, Manassas, VA, USA), 100 U/mL penicillin-streptomycin (Sigma-Aldrich, St. Louis, MO, USA). All cultures were maintained at 37 °C in a 5% CO_2_ humidified incubator to ensure optimal growth conditions. Treatments were conducted under reduced FBS conditions (5%). Apoptosis analysis was performed using annexin V/propidium iodide (V13245, Invitrogen, Carlsbad, CA, USA), dual staining, and evaluated by flow cytometry (BD FACSCalibur, BD Biosciences, San Jose, CA, USA) according to the manufacturer’s protocol. Antibodies against pCRAF (9427), CRAF (12552), PLK1 (4513), MEK1/2 (4694), pMEK1/2 (9121), ERK1/2 (4695), pERK1/2 (4370), E-cadherin (3195), Zeb1 (3396), CD133 (64326), CD44 (3570), NANOG (4903), and GAPDH (5174) were obtained from Cell Signaling Technology (Danvers, MA, USA). Antibodies against TWIST (ab175430) and N-cadherin (ab98952) were acquired from Abcam (Cambridge, MA, USA). Volasertib, a selective inhibitor of PLK1, was procured from Selleck Chemicals (Houston, TX, USA).

### 2.4. Cell Viability Assay

The 10^4^ cells in a 96-well plate were treated with test doses of 5-fluorouracil (5-Fu) and volasertib, alone or in combination, for 48 h in a 0.20 mL volume. Cell viability was measured using the 3-(4,5-dimethylthiazol-2-yl)-2,5-diphenyltetrazolium bromide (MTT) assay, with six replicate wells per dose and vehicle control.

### 2.5. Gene Silencing Using Small Interfering RNA

ERK1/2 siRNA and scrambled control siRNA were sourced from Cell Signaling Technology (Danvers, MA, USA). Transfections were carried out using Lipofectamine 2000 (Invitrogen, Carlsbad, CA, USA) for a duration of 6 h. Subsequently, the lipid–siRNA complex was substituted with fresh culture medium. Cells were then incubated for an additional 48 h before being used for further experimental procedures.

### 2.6. Plasmid and Transfection

PLK1 plasmid DNA and shRNA’s of PLK1 and Zeb1 were sourced from Origene (Rockville, MD, USA). To achieve overexpression of PLK1 and knockdown of PLK1 and Zeb1 in CRC cell lines, Lipofectamine™ 2000 (Invitrogen, Carlsbad, CA, USA) was utilized following the manufacturer’s protocol. Briefly, cells were first seeded in 6-well culture plates. Once the cells attained roughly 50% confluence, transfection was performed using 4 μg of plasmid DNA. Following a 48 h post-transfection interval, stable clones with upregulated PLK1, exhibiting resistance to G418, and stable knockdown clones of PLK1 and Zeb1, demonstrating resistance to puromycin, were isolated. The overexpression and knockdown of PLK1 protein in CRC cells were tested by immunoblotting.

### 2.7. Cell Invasion and Migration Assays 

Cell invasion and migration assays were conducted following standard protocols. Briefly, treated cells were seeded into trans-well inserts, either uncoated for migration or coated with matrigel for invasion, and incubated for 24 h. After incubation, cells were stained using the Diff-Quick stain set and imaged under a microscope.

### 2.8. Sphere-Forming Assay

Cells were seeded at a density of 500 per well in Corning 24-well ultra-low attachment plates (Corning, kennebunk, ME, USA) to facilitate spheroid formation. They were cultured in serum-free DMEM-F12 (ATCC, Manassas, VA, USA) supplemented with B27 (Thermo Fisher Scientific, Waltham, MA, USA), 20 ng/mL EGF (Sigma-Aldrich, St. Louis, MO, USA), 0.4% BSA (Sigma-Aldrich, St. Louis, MO, USA), and 4 μg/mL insulin (Sigma-Aldrich, St. Louis, MO, USA). The medium was changed every two days to maintain optimal growth conditions. On day 14, spheroids larger than 50 μm in diameter were counted and photographed to evaluate their formation and growth, reflecting the self-renewal and stemness characteristics of the cells.

### 2.9. Statistical Analysis

The association between clinicopathological variables and protein expression was analyzed using contingency tables and Chi-square tests. Survival curves were generated using the Kaplan–Meier method, and their significance was determined with the Mantel–Cox log-rank test. Univariate and multivariate logistic regression was used to determine the predictors for lymph node metastasis. The threshold for significance in all analyses was set at a *p*-value of <0.05, with two-sided tests applied for these calculations. Data analysis was conducted using the JMP 14.0 software package (SAS Institute, Inc., Cary, NC, USA).

In all functional assays, the results are expressed as the mean ± standard deviation (SD) derived from triplicate measurements in an independent experiment. Each experiment was conducted a minimum of two times, yielding consistent findings. Statistical significance was assessed using a two-tailed Student’s t-test, with a significance level established at *p* < 0.05.

## 3. Results

### 3.1. PLK-1 Expression in Colorectal Cancer and Its Clinico-Pathological Associations

PLK1 overexpression was noted in 60.3% (693/1149) of CRC cases and was significantly associated with larger tumor size (T3/T4 tumors; *p* = 0.0086), lymph node metastasis (<0.0001), Stage III tumors (*p* = 0.0008), deficient mismatch repair (dMMR) status (*p* = 0.0416). We also found a significant association between PLK1 overexpression and p-ERK1/2 overexpression (*p* < 0.0001) (Figure 1, Table 2). Interestingly, on multivariate logistic regression analysis, we found that PLK1 was an independent predictor of lymph node metastasis (Odds ratio = 1.61, 95% Confidence interval = 1.20–2.17, *p* = 0.0016) (Table 3). However, no association was noted between PLK1 expression and clinical outcomes (DFS, RFS, OS, CSS) (Figure 2).

### 3.2. Inhibition of PLK1 Reverses Chemoresistance in Colorectal Cancer Cells

We found that PLK1 protein expression in our cohort of CRC patients is significantly correlated with aggressive clinicopathological features. This prompted us to explore whether inhibition in PLK1 protein expression in CRC cell lines could reverse chemoresistance. For the in vitro functional studies, initially, we selected two specific CRC cell lines (HT29 and SW480) exhibiting resistance to 5-Fu with high PLK1 expression, along with two other cell lines (DLD1 and HCT116) that were sensitive to 5-Fu with low expressions for PLK1 (Figure 3A,B). Next, we treated these cells with a PLK1-specific inhibitor, volasertib, with and without 5-Fu for 48 h and evaluated the cells for viability and apoptosis. As displayed in Figure 3C,D, inhibition in PLK1 by volasertib significantly increased the chemosensitivity to 5-Fu in both HT29 and SW480 cell lines.

### 3.3. Inhibition of PLK1 Reduces Colorectal Cancer Cell Growth and EMT by Targeting the CRAF-MEK-ERK Signaling Pathway

Previous studies reported that PLK1 promotes cell growth and epithelial–mesenchymal transition through the CRAF-MEK-ERK pathway in various cancers, including gastric cancer [28], breast cancer [29], and prostate cancer [15]. Therefore, we aimed to investigate whether inhibition in PLK1 would reduce colorectal cancer cell growth, EMT, and its role in the CRAF-MEK-ERK signaling pathway. We showed that forced expression of PLK1 promotes CRC cell growth (Figure 4A) and induces the phosphorylation of CRAF, MEK1/2, and ERK1/2 pathway proteins (Figure 4B). Notably, PLK1 knockdown led to a decrease in cell growth, as demonstrated by the clonogenic assay (Figure 4C,D), and reduced the phosphorylation levels of CRAF, MEK1/2, and ERK1/2 proteins. (Figure 4E). We also demonstrated that PLK1 overexpression in DLD1 and HCT116 cell lines increased invasion (Figure 5A, Appendix A) and migration (Figure 5B, Appendix A) and induced epithelial–mesenchymal transition, as evidenced by decreased E-cadherin expression, along with concurrent upregulation in N-cadherin, Zeb1, and Twist (Figure 5C). Conversely, the knockdown of PLK1 decreased invasive (Figure 5D, Appendix A) and migratory potential (Figure 5E, Appendix A) and inhibited the progression of epithelial–mesenchymal transition in HT29 and SW480 cell lines (Figure 5F). These results demonstrate that PLK1 promotes cell growth and epithelial–mesenchymal transition of CRC cell lines, and its inhibition reverses the effect. Moreover, CRAF-MEK-ERK signaling cascade plays an essential role in these events.

### 3.4. Inhibition of PLK1 Decreases the Stemness of Colorectal Cancer Cell

PLK1 exerts a pivotal role in sustaining the characteristics of cancer stemness [29]. To investigate the function of PLK1 in spheroid development and stemness in CRC, we generated PLK1-overexpressing CRC cell lines and grew them in spheroid growth medium. Intriguingly, enforced PLK1 expression in DLD1 and HCT116 cell lines led to enhanced spheroid growth (Figure 6A,B) and an elevated expression of CD133, CD44, and NANOG, compared to empty vector control cells (Figure 6C). Conversely, knockdown of PLK1 resulted in a substantial decrease in spheroid growth (Figure 6D,E) and a decrease in the expression of stem cell markers, including CD133, CD44, and NANOG (Figure 6F).

### 3.5. Knockdown of ERK1/2 Reverses Chemoresistance, Epithelial–Mesenchymal Transition, and Stemness in Colorectal Cancer Cells

To investigate the role of ERK1/2 on chemoresistance, we knockdown ERK1/2 in PLK1-expressing CRC cells, subsequently treating them with 5-Fu for 48 h, and assessed cell viability. Figure 7A shows that ERK1/2 knockdown significantly enhanced the sensitivity of HT29 and SW480 cell lines to 5-Fu treatment, compared to non-specific scrambled siRNA-transfected CRC cells. Knockdown of ERK1/2 significantly decreased the expressions of ERK1/2, N-cadherin, Twist, and Zeb1, accompanied by an elevation in E-cadherin expression (Figure 7B). Additionally, we demonstrated that ERK1/2 knockdown in PLK1-expressing CRC cells reduced stemness, as evidenced by a decrease in spheroid growth (Figure 7C).

### 3.6. Knockdown of Zeb1 Attenuates EMT and Stemness

Previous studies have underscored the crucial role of Zeb1 in PLK1-induced EMT [15], while other reports have underscored its role in regulating stemness [30]. To study the role of Zeb1 in PLK1-induced EMT and stemness, we knocked down Zeb1 in PLK1-expressing CRC cells and analyzed for EMT markers and spheroid growth. Our results revealed that Zeb1 knockdown impeded EMT progression (Figure 8A) and decreased spheroid growth (Figure 8B) in HT29 and SW480 cell lines. These findings underscore the connection between PLK1 and Zeb1, emphasizing their contributions to activating EMT and maintaining stemness in CRC.

## 4. Discussion

Extensive clinical and pathological data proposes that PLK1 signaling plays a key role in tumorigenesis and may serve as a potential biomarker for tumor progression [23,31,32,33,34]. EMT is a pathological process in which epithelial cells lose their characteristic polarity and adhesion properties, undergoing a transformation into a mesenchymal phenotype [35]. The EMT process endows carcinoma cells with increased mobility and invasiveness, serving as a pivotal step in the advancement of cancer metastasis. Despite several studies indicating a correlation between PLK1 expression and the aggressiveness of cancers, including CRC, the specific role of PLK1 in the metastatic progression of CRC and the mechanisms driving PLK1-induced EMT remain under active investigation.

In this study, we initially established that PLK1 is markedly overexpressed in CRC tumors from individuals of Arab descent, with its overexpression associated with several aggressive characteristics, including larger tumor size and an advanced stage. Importantly, multivariate analysis revealed that PLK1 serves as an independent predictor of lymph node metastasis, underscoring its potential role in CRC recurrence and metastasis. Hence, patients showing over-expression of PLK1 may require more aggressive clinical interventions and frequent follow-ups. Similar to our study, Han et al. [36] reported an association between PLK1 over-expression and advanced stage, larger tumor size, and lymphatic metastasis, albeit in a smaller cohort of CRC (*n* = 56). Furthermore, a large meta-analysis [24] including 11 studies (1147 CRC patients) also found a statistically significant association between PLK1 over-expression and adverse clinico-pathological characteristics, such as advanced stage and lymph node metastasis, further corroborating our findings. However, in contrast to our study, this meta-analysis found that CRC patients with PLK1 over-expression had shorter overall survival. This could be attributed to methodological heterogeneity or ethnic differences.

To further explore PLK1′s functional role, we performed a series of in vitro experiments utilizing CRC cell lines. We found that PLK1 promotes colorectal cancer cell metastasis through the MEK/ERK signaling pathway. Our experiment demonstrated that inhibition of PLK1 markedly reduced the migratory and invasive capabilities of CRC cell lines. Furthermore, the downregulation in PLK1 led to the induction of epithelial-like phenotypes, resulting in a transition from mesenchymal to epithelial markers, characterized by an increased E-cadherin and decreased N-cadherin expression.

Our results are consistent with previous studies where similar mechanisms caused prostate cancer progression and metastasis [15]. Several mechanisms of how PLK1 can drive the EMT process in different cancers. For example, in gastric cancer, PLK1 drives EMT via AKT phosphorylation [16]. In non-small-cell lung carcinoma, the involvement of PLK1 in facilitating EMT and metastasis has been linked to the upregulation in the TGF-β/SMAD signaling pathway [17]. Additional research has identified PLK1 as a key driver of FoxM1 activation [31,37]. Limited research has explored the relationship between PLK1 hyperactivity, EMT dysregulation, and drug resistance [9,38]. PLK1 inhibitors, in combination with chemotherapy, have shown enhanced anti-cancer efficacy both in vitro and in vivo by leveraging complementary mechanisms, leading to stronger tumor suppression [39]. Our study also demonstrated that the forced expression of PLK1 induces resistance to 5-Fu in CRC cell lines (Appendix A), while its pharmacological inhibition effectively reverses chemoresistance, restoring sensitivity to 5-Fu treatment.

PLK1 has emerged as a pivotal regulator of the cancer stem cell (CSC) phenotype [40]. Inhibition of PLK1 disrupts the maintenance of stemness in cancer cells, thereby suppressing the stem cell-like traits that contribute to tumorigenesis, therapeutic resistance, and disease progression [29,41,42]. This underscores the therapeutic potential of targeting PLK1 not only to counteract chemoresistance but also to impair CSC-driven tumor regeneration. Our findings demonstrated that the knockdown of PLK1 significantly diminished stem cell-like properties in 5-Fu-resistant CRC cells, implicating PLK1 as a critical driver of stemness in CRC. This suggests that PLK1 contributes to chemoresistance through the induction of CSC characteristics, potentially leading to tumor persistence and relapse. These results emphasize the promise of PLK1 inhibition as a strategy to overcome drug resistance and mitigate stemness-related mechanisms in CRC therapy.

The multifaceted effects of PLK1 inhibition in cancer treatment prompted us to explore the downstream effectors involved. Mechanistically, our study identified the CRAF-MEK-ERK signaling cascade as a downstream target of PLK1. We observed that PLK1 overexpression led to activation of this pathway, while its inhibition significantly suppressed it. These findings are consistent with previous reports in prostate cancer [15], where similar PLK1-mediated regulation of the CRAF-MEK-ERK axis has been demonstrated, further supporting PLK1′s role in modulating key oncogenic pathways across different cancer types. Furthermore, elevated ERK1/2 phosphorylation has been found in 47.3% (543/1149) of CRC patients (Table 2), and inhibition of ERK1/2 suppressed cell growth and induced apoptosis in the CRC cell line [43]. ERK1/2 inhibition notably reversed chemoresistance [44], EMT, and stemness in PLK1-expressing CRC cell lines. Additionally, the knockdown of Zeb1, a mesenchymal transcription factor, significantly reduced EMT and stemness, indicating a potential connection between EMT activation and the maintenance of stem-like properties in CRC. This highlights the key role of the ERK1/2 pathway and Zeb1 in driving both therapeutic resistance and tumor stemness in PLK1-driven tumors.

In our patient cohort, overexpression of ERK1/2 was significantly correlated with elevated PLK1 levels and poor prognosis, underscoring its role in disease progression. Although we did not directly demonstrate that PLK1 regulates ERK1/2 activity, our findings indicate that pharmacological targeting of downstream effectors of PLK1 can overcome 5-Fu resistance, thereby emphasizing the therapeutic potential of ERK1/2 inhibitors in treating PLK1-driven tumors. These results suggest that dual targeting of PLK1 and the ERK1/2 pathway could be an effective strategy to overcome resistance and improve clinical outcomes in CRC.

## 5. Conclusions

In conclusion, our study suggests that targeting PLK1, potentially through its downstream effectors such as ERK1/2, could significantly improve therapeutic outcomes in CRC. The observed association between PLK1 expression, ERK1/2 phosphorylation, and aggressive tumor behavior in our patient cohort further supports the potential of ERK1/2 inhibitors as a therapeutic option in PLK1-driven tumors. This strategy may not only help overcome drug resistance but also inhibit metastatic progression, offering a promising approach for enhancing treatment efficacy in CRC.

## Figures and Tables

**Figure 1 cells-13-01700-f001:**
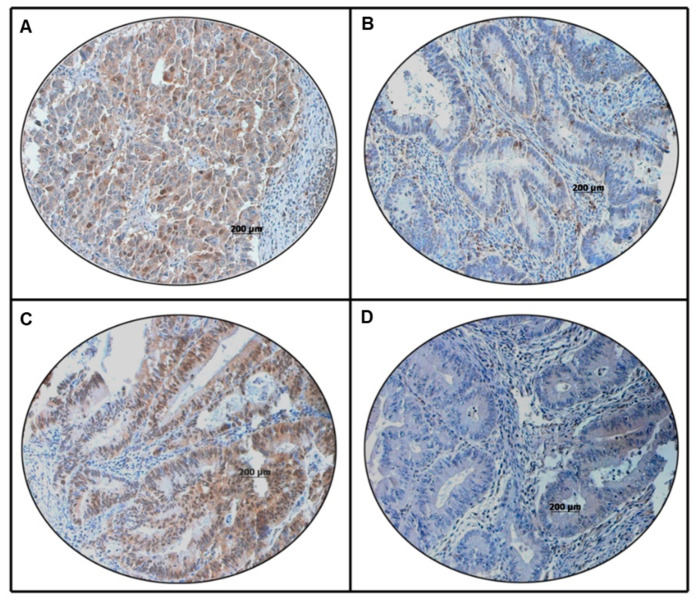
Immunohistochemical analysis of PLK1 and p-ERK1/2 in colorectal cancer (CRC) patients using a tissue microarray (TMA). CRC TMA spots showing overexpression of PLK1 (**A**) and p-ERK1/2 (**C**). In contrast, another set of TMA spots showing reduced expression of PLK1 (**B**) and p-ERK1/2 (**D**). A 20×/0.70 objective on an Olympus BX 51 microscope (Olympus America Inc., Center Valley, PA, USA) was used (Scale bar = 200 μm).

**Figure 2 cells-13-01700-f002:**
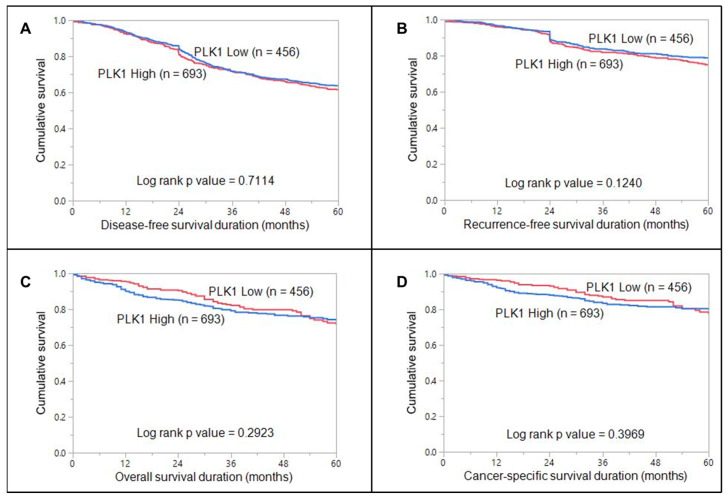
Survival curves for PLK1 in CRC. Kaplan–Meier survival curve showing no significant difference in (**A**) disease-free survival (*p* = 0.7114), (**B**) recurrence-free survival (*p* = 0.1240), (**C**) overall survival (*p* = 0.2923), and (**D**) cancer-specific survival (*p* = 0.3969) were compared between cases with high PLK1 expression and cases with low PLK1 expression.

**Figure 3 cells-13-01700-f003:**
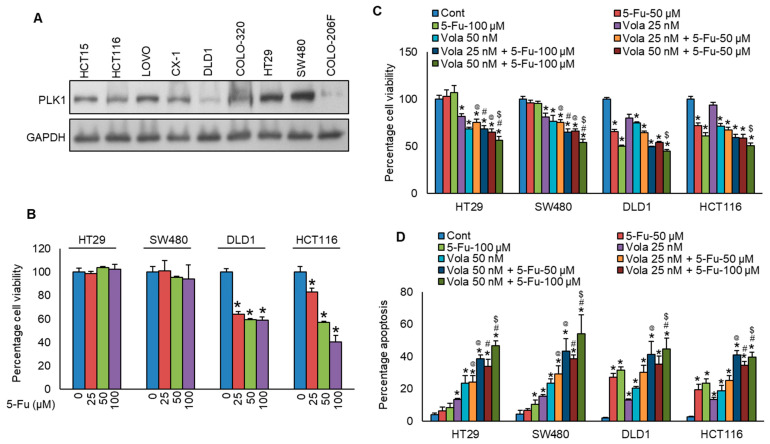
Inhibition in PLK1 reverses chemoresistance in colorectal cancer cells. (**A**) Basal expression levels of PLK1 were assessed in nine CRC cell lines. Proteins were extracted and analyzed via immunoblotting with antibodies against PLK1 and GAPDH. (**B**) CRC cells were exposed to increasing concentrations of 5-fluorouracil (5-Fu) for 48 h, and cell viability was determined by MTT assay (*n* = 6). (**C**,**D**) Volasertib-mediated inhibition in PLK1 reduces chemoresistance in CRC cells, evidenced by decreased cell viability (*n* = 8, (**C**)) and increased induction of apoptosis (*n* = 3, (**D**)). * statistically significant compared to control, @ compared to 5-Fu-50 µM, # compared to 5-Fu-100 µM, $ compared to Vola 50 nM with *p* < 0.05.

**Figure 4 cells-13-01700-f004:**
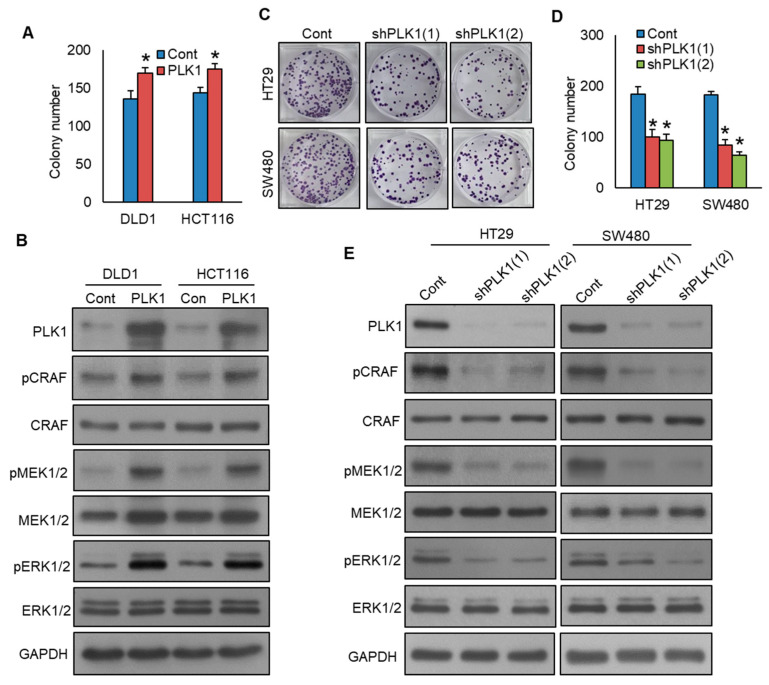
Inhibition of PLK1 reduces colorectal cancer cell growth by targeting the CRAF-MEK-ERK signaling pathway. (**A**) Overexpression of PLK1 increases clonogenicity. The PLK1 overexpressing clones of CRC cells were subjected to clonogenicity assay (*n* = 3). (**B**) Forced expression of PLK1 activates CRAF-MEK-ERK signaling pathway. The proteins from the PLK1-overexpressing clones were analyzed using immunoblotting. (**C**,**D**) Knockdown of PLK1 decreases clonogenicity. CRC cells were transfected with two different shRNA sequences targeting PLK1, and the PLK1 stable knockdown clones of CRC cells were subjected to clonogenicity assay (*n* = 3). (**E**) Knockdown of PLK1 leads to a decrease in the phosphorylation levels of CRAF, MEK1/2, and ERK1/2 proteins. * statistically significant in comparison to the control group, with *p* < 0.05.

**Figure 5 cells-13-01700-f005:**
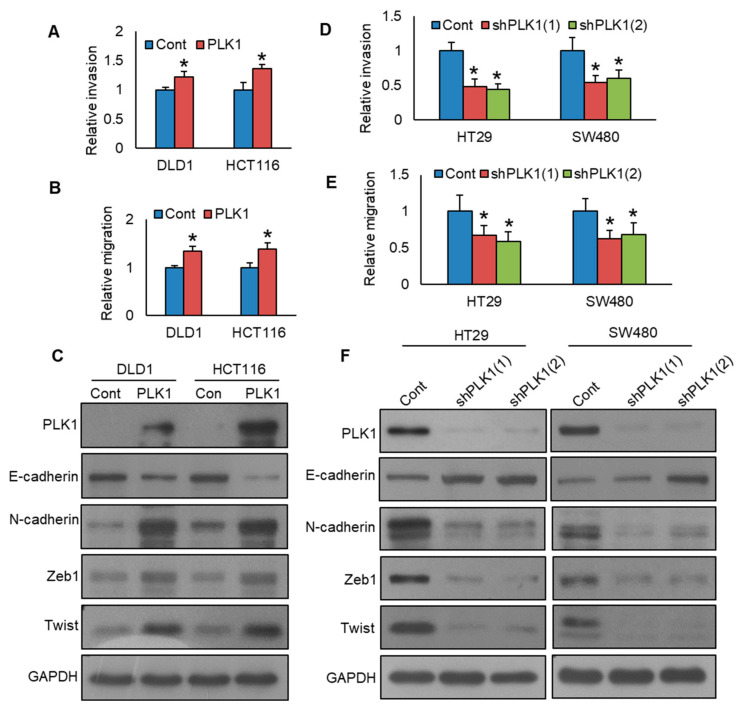
PLK1 inhibition decreases metastatic potential and EMT in colorectal cancer cells. (**A**) Overexpression of PLK1 enhances the invasive ability of CRC cells. CRC cells carrying either an empty vector or PLK1 cDNA were subjected to invasion assay (*n* = 3). (**B**) Overexpression of PLK1 increases the migratory capacity of CRC cells. CRC cells carrying either an empty vector or PLK1 cDNA were subjected to migration assay (*n* = 3). (**C**) Overexpression of PLK1 triggers EMT progression. The proteins isolated from the PLK1 overexpressing CRC clones were immuno-blotted with indicated antibodies (*n* = 3). (**D**) Silencing of PLK1 decreases invasion (*n* = 3). PLK1 stable knockdown CRC cells were subjected to invasion assay (*n* = 3). (**E**) Silencing of PLK1 decreases migration (*n* = 3). PLK1 stable knockdown CRC cells were subjected to migration assay (*n* = 3). (**F**) The silencing of PLK1 leads to a reduction in the expression of EMT markers in CRC cells. The proteins isolated from the PLK1 stable knockdown CRC clones were immuno-blotted with indicated antibodies. * statistically significant in comparison to the control group, with *p* < 0.05.

**Figure 6 cells-13-01700-f006:**
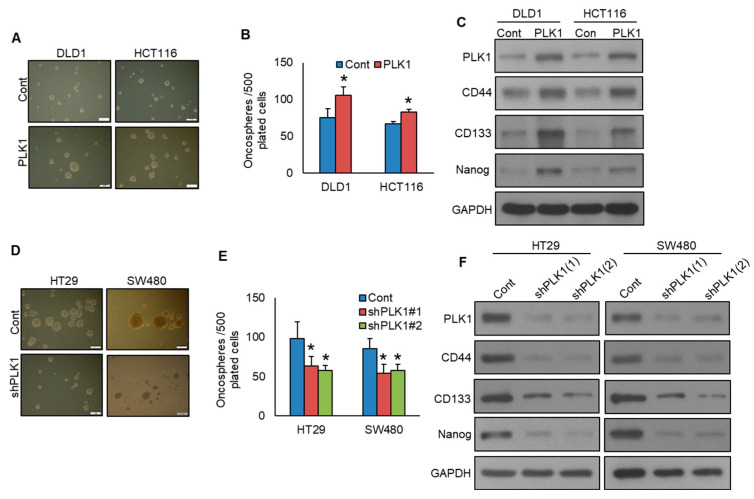
Inhibition of PLK1 decreases the stemness of colorectal cancer cells. (**A**,**B**) Overexpression of PLK1 increases spheroid growth. The CRC clones with PLK1 overexpression were maintained in a sphere-forming medium, and the total quantity of spheroids (>50 μm) in each well was quantified. (*n* = 3) (Scale bar = 1 mm). (**C**) Overexpression of PLK1 enhances stemness, as confirmed by immunoblotting for stem cell markers. (**D**,**E**) Silencing of PLK1 decreases spheroid growth. The PLK1 knockdown CRC clones were subjected to sphere-forming assay (*n* = 3) (Scale bar = 1 mm). (**F**) PLK1 silencing diminishes spheroid stemness, as confirmed by immunoblotting. * statistically significant in comparison to the control group, with *p* < 0.05.

**Figure 7 cells-13-01700-f007:**
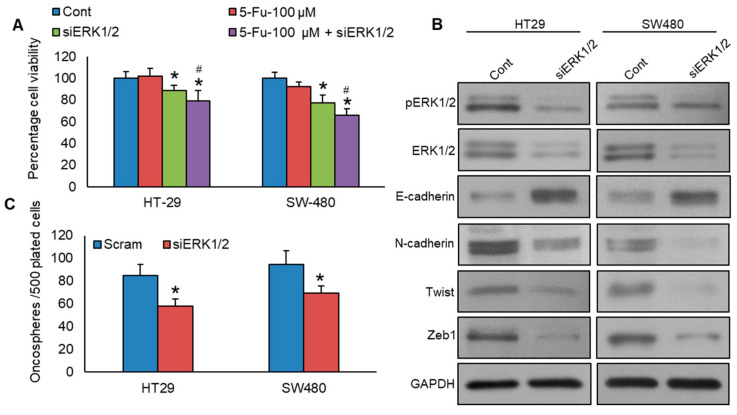
Knockdown of ERK1/2 reverses chemoresistance, EMT, and stemness in colorectal cancer cells. (**A**) Knockdown of ERK1/2 enhances the 5-Fu-induced decrease in cell viability. ERK1/2 knockdown cells were subjected to treatment with or without 5-Fu for a duration of 48 h, and cell viability was determined using the MTT assay (*n* = 8). (**B**) Knockdown of ERK1/2 leads to a decrease in the expression levels of EMT markers in CRC cells. The proteins were isolated from ERK1/2 knockdown CRC cells and were later analyzed using immunoblotting. (**C**) Knockdown of ERK1/2 reduces spheroid growth (*n* = 3). The ERK1/2 knockdown cells grew in a sphere-forming medium, and the total number of spheroids per well was quantified. * statistically significant in comparison to the control group, with *p* < 0.05. # compared to 5-Fu-100 µM.

**Figure 8 cells-13-01700-f008:**
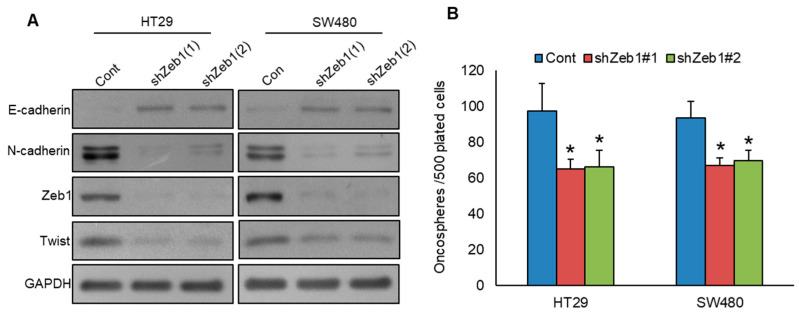
Knockdown of Zeb1 attenuates EMT and stemness. (**A**) Zeb1 knockdown led to a reduction in EMT marker expression in CRC cells. CRC cells were transfected with two different Zeb1 shRNA sequences, and the resulting protein levels in the selected clones were subsequently analyzed via immunoblotting. (**B**) Knockdown of Zeb1 reduces spheroid growth (*n* = 3). CRC clones with Zeb1 knockdown were grown in a specialized sphere-forming medium, and the number of spheroids formed (>50 μm) in each well was carefully counted. * statistically significant compared to control with *p* < 0.05.

**Table 1 cells-13-01700-t001:** Clinicopathological variables for the patient cohort.

	Total	
	n	%
**Total Number of Cases**	1149	
**Age**		
≤50 years	378	32.9
>50 years	771	67.1
**Sex**		
Male	610	53.1
Female	539	46.9
**Tumor Site**		
Left colon	933	81.2
Right colon	216	18.8
**Histological Type**		
Adenocarcinoma	1021	88.9
Mucinous Carcinoma	128	11.1
**pT**		
T1/T2	231	20.1
T3/T4	899	78.2
Unknown	19	1.7
**pN**		
N0	581	50.6
N1/N2	550	47.9
Unknown	18	1.5
**pM**		
M0	995	86.6
M1	145	12.6
Unknown	9	0.8
**TNM Stage**		
I	182	15.9
II	376	32.7
III	446	38.8
IV	145	12.6
**Differentiation**		
Well differentiated	109	9.5
Moderately differentiated	908	79.0
Poorly differentiated	107	9.3
Unknown	25	2.2
**MMR status**		
dMMR	108	9.4
pMMR	1041	90.6

pT—pathological tumor stage, pN—pathological nodal stage, pM—pathological metastasis stage, TNM—tumor, node, metastasis; dMMR—deficient mismatch repair; pMMR—proficient mismatch repair.

**Table 2 cells-13-01700-t002:** Correlation of PLK1 immunohistochemical expression with clinico-pathological parameters in colorectal carcinoma.

	PLK1 Low		PLK1 High		*p* Value
	n	%	n	%	
**Total Number of Cases**	456	36.7	693	60.3	
**Age**					
≤50 years	142	31.1	236	34.1	0.3036
>50 years	314	68.9	457	65.9	
**Sex**					
Male	249	54.6	361	52.1	0.4037
Female	207	45.4	332	47.9	
**Tumour Site**					
Left colon	377	82.7	556	80.2	0.2994
Right colon	79	17.3	137	19.8	
**Histological Type**					
Adenocarcinoma	395	86.6	626	90.3	0.0506
Mucinous Carcinoma	61	13.4	67	9.7	
**pT**					
T1/T2	109	24.3	122	17.9	0.0086
T3/T4	339	75.7	560	82.1	
**pN**					
N0	265	59.1	316	46.3	<0.0001
N1/N2	183	40.9	367	53.7	
**pM**					
M0	395	87.6	600	87.1	0.8042
M1	56	12.4	89	12.9	
**TNM Stage**					
I	90	19.7	92	13.3	0.0008
II	162	35.5	214	30.9	
III	148	32.5	297	42.9	
IV	56	12.3	89	13.0	
**Differentiation**					
Well differentiated	47	10.6	62	9.1	0.4191
Moderately differentiated	351	78.9	557	82.0	
Poorly differentiated	47	10.6	60	8.8	
**MMR status**					
dMMR	33	7.2	75	10.8	0.0416
pMMR	423	92.8	618	89.2	
**p-ERK1/2 IHC**					
Low (H score < 60)	313	68.6	293	42.3	<0.0001
High (H score ≥ 60)	143	31.4	400	57.7	

pT—pathological tumor stage, pN—pathological nodal stage, pM—pathological metastasis stage, TNM—tumor, node, metastasis; dMMR—deficient mismatch repair; pMMR—proficient mismatch repair.

**Table 3 cells-13-01700-t003:** Univariate and multivariate analysis for predictors of lymph node metastasis using logistic regression.

	Lymph Node Metastasis			
	Univariate		Multivariate	
	Odds Ratio (95% CI)	*p* Value	Odds Ratio (95% CI)	*p* Value
**Age** (>50 vs. ≤50 years)	0.69 (0.54–0.89)	0.0043	0.69 (0.51–0.94)	0.0201
**Sex** (Male vs. Female)	0.92 (0.72–1.16)	0.4571		
**Tumor site** (Left vs. Right)	1.25 (0.93–1.69)	0.1427		
**Histologic type** (Mucinous vs. Adenocarcinoma)	1.23 (0.85–1.78)	0.2795		
**pT** (T3/4 vs. T1/2)	4.79 (3.40–6.76)	<0.0001	3.44 (2.22–5.32)	<0.0001
**Stage** (IV vs. I–III)	3.44 (2.30–5.16)	<0.0001	3.11 (2.05–4.72)	<0.0001
**Grade** (3 vs. 1–2)	1.79 (1.18–2.70)	0.0061	1.56 (0.98–2.49)	0.0619
**PLK1** (High vs. Low)	1.68 (1.32–2.14)	<0.0001	1.61 (1.20–2.17)	0.0016

pT—pathological tumor stage.

## Data Availability

All data generated or analyzed during this study are included in this published article.

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
