# Peer review of "Polo-like Kinase 1 Predicts Lymph Node Metastasis in Middle Eastern Colorectal Cancer Patients; Its Inhibition Reverses 5-Fu Resistance in Colorectal Cancer Cells"

_cells, 2024, doi:10.3390/cells13201700_

Round 1

Reviewer 1 Report

Comments and Suggestions for Authors

In this study, authors investigated PLK1 as a marker of invasiveness and drug resistance in colorectal cancer. The authors employed a wide range of experiments, from correlation assessments of PLK1 expression in tumor samples with clinicopathological findings to in vitro studies using gene silencing and gene overexpression in CRC cells. They also conducted functional assays to assess cell viability, migration, invasion, drug sensitivity, and tumorsphere formation. Furthermore, the authors expanded their investigations to explore mechanisms underlying their functional findings, particularly focusing on MAPK/Erk pathway activation and EMT markers. The authors have concluded that PLK1 is a marker of CRC lymph node metastasis, it is associated with EMT and Erk activation and is related to CRC cell resistance to 5-Fu.  

While this research is not entirely original, as PLK1 has been studied in various malignancies and CRC, the focus on Arab ethnicity and the expansion of PLK1 association with EMT are interesting aspects of the current study. The comprehensive experimentation reported here adds value to this work. However, there are some concerns that require the authors' attention.

Methods: A clearer description of the methods is needed. For instance, your migration and invasion assays are not explained in the method section. Also, did you apply any criteria to select or exclude patient samples? The MTT assay and Zeb shRNAs are not mentioned in the methods section either. For the sphere-forming assay, please clarify whether you considered the size of spheres in your counting. If two groups have the same number of spheres but the spheres in one group are larger, this indicates a difference between the groups that needs to be addressed.

Results: Please define all abbreviations in your table legends. For Figure 2, there is no “n” number provided for the comparisons. Is it the same as the numbers shown in the tables?

For Figure 3, since PLK1 inhibitor (Vola.) alone significantly reduces cell viability, the difference between Vola 50 nM and Vola 50 nM + 5-Fu-100 µM should be statistically tested, otherwise, the combined effect may be solely attributed to Vola. The same applies to the apoptosis assay.

For Figure 5, there is no information or images provided for the migration and invasion assays.

For Figure 6, panel D (SW480 images), there appear to be minimal differences in the number of tumorspheres between the groups. Please clarify in the methods section how the size of tumorspheres was factored into your analysis.

Discussion: The discussion section requires significant improvement, as it currently mainly reiterates the results. Since a major focus of your manuscript is the regulation of EMT by PLK1, I would expect a more comprehensive discussion of how this association exists. Please elaborate on this in more detail with reference to both your own findings and the literature.

Also, discuss how PLK1 is associated with drug resistance. Since PLK1 also affects cell viability and apoptosis, what potential mechanisms are involved based on your study?

Additionally, the potential clinical implications of your work should be better addressed. Have PLK1 or Erk inhibitors been studied in clinical trials for CRC or other cancers? What are the benefits and potential challenges? Your data does not show any differences in patient survival based on PLK1 low or high status. How do you interpret this for future clinical interventions? Do you observe any differences in patient survival based on Erk activation status?

 Again, this is a strong study supported by extensive experimentation, but some improvements are needed. I hope you find these comments helpful.

Comments on the Quality of English Language

Overall, the text reads very well. 

I noticed minor error, eg. Line 64, I think "poor prognosis" would be correct.

line 76, Methods subsection title is Sample Section which I think is "Selection"

line 336, discussion, "We found that PLK1 promotes colorectal cancer cell metastasis through.." This is not accurate, you tested cell migration and invasion not metastasis.

Author Response

Reviewer 1

While this research is not entirely original, as PLK1 has been studied in various malignancies and CRC, the focus on Arab ethnicity and the expansion of PLK1 association with EMT are interesting aspects of the current study. The comprehensive experimentation reported here adds value to this work. However, there are some concerns that require the authors' attention.

We are thankful to the reviewer for their valuable time to review our manuscript and for their appreciative comments. It is indeed our endeavor to see our results being translated into pre-clinical and clinical trials for the benefit of the patients. We thank you once again for your positive review. The concern of the reviewer is addressed below.

Methods: A clearer description of the methods is needed. For instance, your migration and invasion assays are not explained in the method section. Also, did you apply any criteria to select or exclude patient samples? The MTT assay and Zeb shRNAs are not mentioned in the methods section either. For the sphere-forming assay, please clarify whether you considered the size of spheres in your counting. If two groups have the same number of spheres but the spheres in one group are larger, this indicates a difference between the groups that needs to be addressed.

Thank you for your valuable feedback. We have revised the methods section to provide a clearer description of the assays and criteria used. Specifically:

Migration and Invasion Assays: We have now included a detailed explanation of the protocols used for these assays in the methods section.

Patient Sample Criteria: We would like to clarify that no inclusion or exclusion criteria were applied for patient sample selection.

MTT Assay and Zeb shRNAs: The methods for the MTT assay and Zeb shRNA knockdown have been added to the methods section.

Sphere-Forming Assay: We have clarified the analysis of tumorspheres in the methods section as follows: “Spheroids larger than 50 μm were counted and photographed”.

We believe these revisions address your concerns and provide a more comprehensive overview of our experimental procedures.

Results: Please define all abbreviations in your table legends. For Figure 2, there is no “n” number provided for the comparisons. Is it the same as the numbers shown in the tables?

We thank the reviewer for their suggestion to define all abbreviations in our table legends. As suggested by the reviewer, we have now defined the abbreviations in the table legends. We would also like to clarify that numbers for PLK1 low and high groups for figure 2 are the same as the numbers shown in the tables. We have now incorporated the “n” number in Figure 2 for comparisons.

For Figure 3, since PLK1 inhibitor (Vola.) alone significantly reduces cell viability, the difference between Vola 50 nM and Vola 50 nM + 5-Fu-100 µM should be statistically tested, otherwise, the combined effect may be solely attributed to Vola. The same applies to the apoptosis assay.

Thank you for pointing this out. We have tested the statistical difference between Vola alone and the combination treatment, and the results have been updated in the manuscript. These additional analyses clarify the contribution of each treatment to the observed effects, as suggested. Thank you for the valuable input!

For Figure 5, there is no information or images provided for the migration and invasion assays.

Thank you for your observation. We have now revised the manuscript and included the images for the migration and invasion assays as Figure S1.

For Figure 6, panel D (SW480 images), there appear to be minimal differences in the number of tumorspheres between the groups. Please clarify in the methods section how the size of tumorspheres was factored into your analysis.

Thank you for your insightful comment regarding Figure 6, panel D. We measured the diameter of each tumorsphere to assess their size, which provides additional context for comparing the groups. We have clarified the analysis of tumorspheres in the methods section as follows: “Spheroids larger than 50 μm were counted and photographed”.

Discussion: The discussion section requires significant improvement, as it currently mainly reiterates the results. Since a major focus of your manuscript is the regulation of EMT by PLK1, I would expect a more comprehensive discussion of how this association exists. Please elaborate on this in more detail with reference to both your own findings and the literature.

Thank you for your valuable feedback regarding the discussion section. We have made significant revisions to improve its quality and depth. We have elaborated on our findings and integrated relevant literature to provide a more comprehensive analysis. All changes have been highlighted in red in the manuscript for your convenience.

Also, discuss how PLK1 is associated with drug resistance. Since PLK1 also affects cell viability and apoptosis, what potential mechanisms are involved based on your study?

Thank you for your valuable comment. Our data clearly demonstrate that PLK1 is overexpressed in the chemoresistant cell lines HT29 and SW480, while its expression is low in HCT116 and DLD1. Furthermore, forced expression of PLK1 activates the CRAF-MEK-ERK signaling pathway, and its inhibition reverses this activation. Additionally, we have now showed that PLK1 overexpression promotes cell proliferation and induces chemoresistance in these cell lines (Figure S2).

Additionally, the potential clinical implications of your work should be better addressed. Have PLK1 or Erk inhibitors been studied in clinical trials for CRC or other cancers? What are the benefits and potential challenges? Your data does not show any differences in patient survival based on PLK1 low or high status. How do you interpret this for future clinical interventions? Do you observe any differences in patient survival based on Erk activation status?

We acknowledge the reviewer’s concern regarding lack of differences in patient survival based on PLK1 low or high status and its impact on clinical interventions. Although PLK1 expression was not associated with patient prognosis, we found that its over-expression was significantly associated with other adverse clinico-pathological characteristics such as larger tumor size, advanced stage and lymph node metastasis (for which PLK1 over-expression was an independent predictor). These findings suggest that PLK1 may have a potential role in tumor recurrence and metastasis. Hence, patients showing over-expression of PLK1 may require more aggressive clinical interventions and frequent follow-ups. We have now incorporated this in the Discussion section of the revised manuscript.

We thank the reviewer for their suggestion to look for any differences in patient survival based on Erk activation status. We found that p-ERK1/2 over-expression was associated with overall survival on Kaplan-Meier curve analysis. However, no difference was noted between p-ERK1/2 expression and disease-free survival, recurrence-free survival or cancer-specific survival. We have provided the survival curves below for reviewer’s concern only.

Figure: Survival curves for p-ERK1/2 in CRC.  Kaplan Meier survival curve showing no significant difference in (A) disease-free survival (p = 0.0780), (B) recurrence-free survival (p = 0.7228) and (D) cancer-specific survival (p = 0.0863) between p-ERK1/2  high expression cases and p-ERK1/2 low expression cases, but (C) poor overall survival (p = 0.0096) in p-ERK1/2 high cases compared to p-ERK1/2 low cases.

I noticed minor error, eg. Line 64, I think "poor prognosis" would be correct.

Thank you for bringing this to our attention. We have corrected the minor error in line 64, changing "poor prognosis" to ensure accuracy. We appreciate your careful review of the manuscript!

line 76, Methods subsection title is Sample Section which I think is "Selection"

Thank you for your observation. We have corrected the subsection title in line 76 from "Sample Section" to "Sample Selection" to accurately reflect its content. We appreciate your attention to detail!

line 336, discussion, "We found that PLK1 promotes colorectal cancer cell metastasis through.." This is not accurate, you tested cell migration and invasion not metastasis.

Thank you for highlighting this inaccuracy. We have revised the statement in line 336 of the discussion to clarify that our study focused on testing metastatic potential, as cell migration and invasion serve as indicators of this potential rather than confirming actual metastasis. The updated sentence now accurately reflects our findings. We appreciate your thorough review and valuable feedback!

Once again, we are really gratified and thankful for the reviewer’s efforts and for the opportunity to refine our manuscript further, for consideration for publication in Molecular Oncology. We hope that our response to the reviewers and the amendments in the manuscript are to the satisfaction of the editorial team and that you now deem the manuscript worthy of publication as a research article.

Thanking you

Sincerely Yours

Khawla S. Al-Kuraya, MD, FCAP

Director of Research

Human Cancer Genomic Research,

King Faisal Specialist Hospital and Research Cancer

MBC#98-16, P.O. Box 3354, Riyadh 11211, Saudi Arabia.

Email: kkuraya@kfshrc.edu.sa

Reviewer 2 Report

Comments and Suggestions for Authors

This study suggests that PLK1 may be a predictive marker for lymphnode metastasis in Middle Eastern colorectal cancer patients. The authors also reported that PLK1 could alleviate 5-FU resistance in colorectal cancer cells. The authors also suggested that PLK1 may induce EMT and stemness in colorectal cancer by associating with ERK1/2 and Zeb1. This study reported the potential role of PLK1 in colorectal cancer in various aspects, but several points should be considered.

-       A key priority for improvement is bridging the gap between studies in clinical samples and in vitro studies. The novelty of this study in clinical samples is the role of PLK1 in predicting lymph node metastasis in patients with colorectal cancer in the Middle East. What are the similarities and differences between the data from this patient cohort study and previous cohort studies conducted in other regions? How does the in vitro study relate to the uniqueness of this study, which analyzed samples from Middle Eastern patients?

-       Clinical studies have suggested a role for PLK1, focusing on lymph node metastasis. However, in vitro studies have suggested a role for PLK1, focusing on improving 5-FU resistance. Although the analysis of markers for EMT may be related to metastasis, more in-depth and precise experimental designs are needed. As these gaps between clinical and in vitro research topics are filled, the title of this manuscript may become more refined.

-       Apart from the data on cell viability, it is not clear from this study how 5-FU modulates several physiological characteristics and signaling pathways in colorectal cancer.

-       The Discussion section of this manuscript clearly needs improvement. It should clearly state what is new and meaningful about this study and what are its limitations. It also needs a more in-depth discussion of the data presented in this manuscript. In particular, the lack of description of EMT, stemness, and Zeb1 protein in colorectal cancer and how these relate to the results of the cohort study should be described.

-       I am not sure how invasion and migration were analyzed in this study. Can microscope images be included?

-       The use of terms and abbreviations in this manuscript should be reviewed more carefully. For example, terms such as '5-Fu', '5-FU', 'FU', and 'Fu' should be unified.

Author Response

Reviewer 2

This study suggests that PLK1 may be a predictive marker for lymphnode metastasis in Middle Eastern colorectal cancer patients. The authors also reported that PLK1 could alleviate 5-FU resistance in colorectal cancer cells. The authors also suggested that PLK1 may induce EMT and stemness in colorectal cancer by associating with ERK1/2 and Zeb1. This study reported the potential role of PLK1 in colorectal cancer in various aspects, but several points should be considered.

We would like to express our gratitude to the reviewer for their time and effort in reviewing our manuscript, as well as for their positive feedback and insightful comments. We have addressed the reviewer’s concerns as follows.

-       A key priority for improvement is bridging the gap between studies in clinical samples and in vitro studies. The novelty of this study in clinical samples is the role of PLK1 in predicting lymph node metastasis in patients with colorectal cancer in the Middle East. What are the similarities and differences between the data from this patient cohort study and previous cohort studies conducted in other regions? How does the in vitro study relate to the uniqueness of this study, which analyzed samples from Middle Eastern patients?

We thank the reviewer for their suggestion to look for similarities and differences between the data from this patient cohort study and previous cohort studies conducted in other regions with respect to clinicO-pathological associations of PLK1. We have now incorpoprated this information in the Discussion section a follows: “Similar to our sudy, Han et al. (PMID:22648245; PMCID:PMC3560731) reported an association between PLK1 over-expression and advanced stage, larger tumor size and lymphatic matastasis, albeit in a smaller cohort of CRC (n = 56). Furthermore, a large meta-analysis (https://doi.org/10.1016/j.gene.2019.144097) including 11 studies (1147 CRC patients) also found a statistically significant association between PLK1 over-expression and adverse clinico-pahological characteristics such as advanced stage and lymph node metastasis, further corroborating our findings. However, in contrast to our study, this meta-analysis found that CRC patients with PLK1 over-expression had shorter overall survival. This could be attributed to methodological heterogeneity or ethnic differences.”

-       Clinical studies have suggested a role for PLK1, focusing on lymph node metastasis. However, in vitro studies have suggested a role for PLK1, focusing on improving 5-FU resistance. Although the analysis of markers for EMT may be related to metastasis, more in-depth and precise experimental designs are needed. As these gaps between clinical and in vitro research topics are filled, the title of this manuscript may become more refined.

Thank you for your thoughtful comments. We acknowledge the reviewer's suggestion that more in-depth and precise experimental designs are needed. In addition to our in vitro chemoresistance experiments, we have also studied and demonstrated the effects of PLK1 and its inhibition on EMT markers, as well as the migration and invasion potential of colorectal cancer cell lines. These factors are important indicators of metastatic potential, providing further insights into PLK1's role in metastasis.

-       Apart from the data on cell viability, it is not clear from this study how 5-Fu modulates several physiological characteristics and signaling pathways in colorectal cancer.

Thank you for your insightful comment. Our study demonstrated that forced expression of PLK1 activates the CRAF-MEK-ERK signaling pathway in the chemosensitive cell lines DLD1 and HCT116, while knockdown of PLK1 in the chemoresistant cell lines HT29 and SW480 leads to the downregulation of this pathway. Additionally, we observed that PLK1 overexpression promotes cell proliferation and induces chemoresistance in these cell lines (Figure S2), further supporting its role in modulating these pathways. However, we agree that a more comprehensive investigation into how 5-Fu modulates various physiological characteristics and signaling pathways in colorectal cancer is crucial. In future studies, we plan to explore these aspects in greater detail to provide a more complete understanding of 5-Fu's mechanisms of action beyond cell viability. We appreciate your valuable feedback and will consider this in our ongoing research.

-       The Discussion section of this manuscript clearly needs improvement. It should clearly state what is new and meaningful about this study and what are its limitations. It also needs a more in-depth discussion of the data presented in this manuscript. In particular, the lack of description of EMT, stemness, and Zeb1 protein in colorectal cancer and how these relate to the results of the cohort study should be described.

Thank you for your valuable feedback. We acknowledge that the Discussion section required improvement, and we have revised and expanded it to clearly highlight the novel and meaningful contributions of our study, along with its limitations. We also discussed a more in-depth analysis of the presented data, with particular focus on the role of PLK1 on EMT, stemness, and Zeb1 protein in CRC progression and therapeutic resistance based on our own findings and previous literature.

-       I am not sure how invasion and migration were analyzed in this study. Can microscope images be included?

Thank you for your comment. We have now incorporated the microscope images of invasion and migration as Figure S1.

 -       The use of terms and abbreviations in this manuscript should be reviewed more carefully. For example, terms such as '5-Fu', '5-FU', 'FU', and 'Fu' should be unified.

Thank you for your comment and appreciate your attention to detail. We have carefully reviewed the manuscript and standardized the use of terms and abbreviations, ensuring consistency throughout. Specifically, all references to 5-Fluorouracil have been unified as '5-Fu'.

Once again, we are really gratified and thankful for the reviewer’s efforts and for the opportunity to refine our manuscript further, for consideration for publication in Molecular Oncology. We hope that our response to the reviewers and the amendments in the manuscript are to the satisfaction of the editorial team and that you now deem the manuscript worthy of publication as a research article.

Thanking you

Sincerely Yours

Khawla S. Al-Kuraya, MD, FCAP

Director of Research

Human Cancer Genomic Research,

King Faisal Specialist Hospital and Research Cancer

MBC#98-16, P.O. Box 3354, Riyadh 11211, Saudi Arabia.

Email: kkuraya@kfshrc.edu.sa

Round 2

Reviewer 1 Report

Comments and Suggestions for Authors

Authors have addressed my comments, and the revised manuscript is improved and, in my opinion, suitable for publication. 

Reviewer 2 Report

Comments and Suggestions for Authors

I think the authors have adequately responded to the comments presented.